# Transmission dynamics of *Klebsiella pneumoniae* in a neonatal intensive care unit in Zambia before and after an infection control bundle

Laura T. Phillips[1]*, Matthew Bates[2,3], Susan E. Coffin[4,5], Ebenezer Foster-Nyarko[1], Monica Kapasa[6], Sylvia Machona[7], Lawrence Mwananyanda[8], James C. L. Mwansa[9], Chileshe L. Musyani[10,11], John M. Tembo[3], Franklyn N. Egbe[12], Kathryn E. Holt[1,13c], Davidson H. Hamer[14,15,16c]

1 Department of Infection Biology, London School of Hygiene & Tropical Medicine, London, United Kingdom, 2 School of Natural Sciences, University of Lincoln, Lincoln, United Kingdom, 3 HerpeZ, University Teaching Hospital, Lusaka, Zambia, 4 Department of Pediatrics, Perelman School of Medicine, University of Pennsylvania, Philadelphia, Pennsylvania, United States of America, 5 Division of Infectious Diseases, Children's Hospital of Philadelphia, Philadelphia, Pennsylvania, United States of America, 6 Department of Pediatrics, University Teaching Hospital, Lusaka, Zambia, 7 Neonatology Department, University Teaching Hospital, Lusaka, Zambia, 8 Biomedical Department, Right to Care, Lusaka, Zambia, 9 Directorate of Research and Post Graduate Studies, Lusaka Apex Medical University, Lusaka, Zambia, 10 Institute of Basic and Biomedical Sciences, Levy Mwanawasa Medical University, Lusaka, Zambia, 11 Antimicrobial Resistance Unit, Zambia National Public Health Institute, Lusaka, Zambia, 12 Institute of Infection, Veterinary & Ecological Sciences, University of Liverpool, Liverpool, United Kingdom, 13 Department of Infectious Diseases, School of Translational Medicine, Monash University, Melbourne, Victoria, Australia, 14 Department of Global Health, Boston University School of Public Health, Boston, Massachusetts, United States of America, 15 Section of Infectious Diseases, Boston University Chobanian & Avedisian School of Medicine, Boston, Massachusetts, United States of America, 16 Boston University Center on Emerging Infectious Diseases, Boston, Massachusetts, United States of America

☯ These authors contributed equally to this work.

* Laura.Phillips@lshtm.ac.uk

## Abstract

*Klebsiella pneumoniae* is a leading cause of neonatal sepsis in low- and middle-income countries, with antimicrobial resistance (AMR) significantly contributing to mortality. We used whole genome sequencing to explore the impact of an infection prevention and control (IPC) intervention on *K. pneumoniae* strains and transmission dynamics responsible for sepsis in a Zambian neonatal unit. Blood culture isolates were collected during the Sepsis Prevention in Neonates in Zambia (SPINZ) study, including a 7-month baseline period and 12 months following implementation of a low-cost IPC bundle. *K. pneumoniae* genomes associated with 411 neonatal infections were characterised, comprising 24 unique sequence types (STs) and dominated by ST307 (69.3%, n = 285). Nearly all isolates (99.0%) carried extended spectrum beta-lactamases, but few carried carbapenemases (2.7%). Most infections (95.6%) were associated with probable transmission clusters, ranging in size from 2–202 patients and spanning durations of 2–232 days. Most *K. pneumoniae* (n = 228, 70%) were isolated during the 7-month baseline period and formed six clusters, including one cluster of >200 neonates infected with ST307. Transmission of all strains was

**Data availability statement:** All analysis and visualisation code are available at https://github.com/klebgenomics/SPINZ (DOI: https://doi.org/10.5281/zenodo.17589084). Whole-genome sequence data were deposited by the sequencing laboratory (Quadram Institute of Bioscience, UK) under BioProject PRJEB46513.

**Funding:** The SPINZ study was funded by the Thrasher Research Fund [https://www.thrasherresearch.org/] (grant #12036 to DHH). Support for the sequencing and analysis of K. pneumoniae was provided by the Gates Foundation [https://www.gatesfoundation.org/] (INV005691 to DHH, INV069410 and INV077266 to KEH). The study was designed, analyzed and implemented by the authors. The funders had no role in study design, data collection and analysis, decision to publish, or preparation of the manuscript. The conclusions and opinions expressed in this work are those of the author(s) alone and shall not be attributed to the funders. Under the grant conditions of the Gates Foundation, a Creative Commons Attribution 4.0 License has already been assigned to the Author Accepted Manuscript version that might arise from this submission. Please note works submitted as a preprint have not undergone a peer review process.

**Competing interests:** The authors have declared that no competing interests exist.

periodically suppressed by an IPC bundle; however not all strains were eliminated, and some were able to re-emerge later to re-establish infection and transmission, alongside newly introduced strains that formed additional transmission clusters. Some clusters were associated with rapid onset of disease (within 2 days of admission) and others with delayed onset, suggesting different sources of contamination (e.g., reagent vs environmental). These findings reinforce the need for sustained IPC efforts, and better understanding of environmental reservoirs of opportunistic pathogens in neonatal units to inform such efforts.

## Introduction

In 2019, an estimated 2.44 million neonatal deaths were attributable to infectious causes [1], with 99% of these deaths occurring in low- and middle-income countries (LMICs) [2]. Neonatal sepsis in LMICs contributes significantly to this high burden, accounting for 78.9% of the world's total reported neonatal sepsis cases and 93.9% of global neonatal sepsis-related deaths in 2019 [3–5]. *Klebsiella* species, predominantly *Klebsiella pneumoniae*, are a leading agent of neonatal sepsis [4,6], with infections additionally often resistant to World Health Organization (WHO) recommended treatment regimens [7]. It is estimated that the fraction of sepsis deaths attributable to antimicrobial resistance (AMR) increased by 18% in children younger than 5 years between 1990–2019, with *K. pneumoniae* being one of the biggest contributors to AMR-attributable sepsis deaths in 2021 [8,9]. Of particular concern is the increasing proportion of neonatal deaths attributable to extended-spectrum beta-lactamase (ESBL) producing or carbapenem-resistant *K. pneumoniae*, which was declared a critical priority in the WHO Bacterial Priority Pathogen List [8,10].

Accurate and timely diagnosis of neonatal sepsis in LMICs is challenging, due to shared clinical features with many common conditions, and neonates not consistently presenting with severe clinical signs [11,12]. Additionally, limited availability of blood culture and other laboratory testing may limit accurate and prompt diagnosis, ultimately leading to underestimation of the burden of neonatal sepsis [12].

Given the prevalence of AMR among *Klebsiella* isolates, the growing burden of *K. pneumoniae* related sepsis, and the diagnostic challenges, there is a renewed focus on prevention. A significant measure proposed to reduce the burden of neonatal sepsis is the introduction of a maternal vaccine, targeting the external capsular (K) and/or lipopolysaccharide (O) antigens of *K. pneumoniae* [13,14]. Pathogen-agnostic prevention measures, such as infection prevention and control (IPC) interventions within neonatal intensive care units (NICUs), are more realistic and achievable in the short-term, with some having been shown to be effective in reducing infection and mortality. However, the evidence base is limited, and the relative efficacy of different intervention strategies is not yet clear [15–18].

The impact of prevention strategies on pathogen populations can be informed by pathogen whole genome sequencing (WGS). Evaluations of IPC interventions measure key outcomes such as the number of infections and fatalities; however,

pathogen WGS can be additionally used to enhance our understanding of the underlying bacterial populations targeted by the intervention, how they change in response to the intervention, and whether changes are stable over the duration of the intervention [19]. In the absence of serological K typing, which is only available in one centre globally, WGS is also the predominant tool for profiling K and O serotypes in *K. pneumoniae* [20,21].

We recently reported an evaluation of an IPC intervention in a NICU in Zambia, where the majority of positive blood cultures (>70%) yielded *K. pneumoniae* [22]. The multi-faceted intervention included IPC training, text message reminders, alcohol hand rub, enhanced environmental cleaning, and weekly bathing of babies ≥1.5 kg with 2% chlorhexidine gluconate. Mortality decreased following the implementation of these measures compared with the pre-intervention (baseline) period of observation [22]. Here we use WGS to characterize stored *K. pneumoniae* clinical isolates from that study, to compare the populations before and after the IPC intervention and investigate its impact on the pathogen population.

## Methods

### Ethics

The Sepsis Prevention in Neonates in Zambia (SPINZ) study was reviewed and approved by the institutional review boards and ethics boards at Boston University, Children's Hospital of Philadelphia, and the Excellence in Research Ethics and Science (ERES) Converge in Zambia. Written informed consent was provided by the mothers of neonates recruited into the study. Retrospective investigation of transmission using WGS was also approved by the Observational/Interventions Research Ethics Committee of the London School of Hygiene and Tropical Medicine (ref #29931). For the purpose of the genomics investigation reported in this manuscript, fully anonymised data were accessed from the 5th of June 2024, and none of the authors had access to information that could identify individual participants.

### Study population

SPINZ was a prospective observational cohort study of hospitalized neonates conducted in a large tertiary care University Teaching Hospital (UTH) in Lusaka, Zambia. Full details of the original intervention study were reported previously [22].

Briefly, a 6-month baseline period ('Baseline'), was followed by six weeks of IPC bundle implementation and 11 months of intervention assessment, both combined in this study to be referred to as the 'Post-implementation' period. During SPINZ, neonates admitted to the NICU between September 2015, and March 2017 were enrolled in the prospective study, and blood cultures were obtained from all neonates with clinically suspected sepsis (defined as fever, hypothermia, tachycardia or bradycardia, hypoglycemia, respiratory difficulty, new-onset seizures, lethargy, poor feeding, abdominal distention, vomiting, diarrhea, or poor perfusion) [23]. The NICU has approximately 3300 admissions per year and is typically filled beyond its official capacity (60 cots and an average daily count of 75 infants). Blood culture was performed at the UTH microbiology laboratory, Lusaka, Zambia using BD Bactec FX200 culture system (BD Life Sciences, New Jersey, USA), followed by species identification and antibiotic susceptibility testing with the automated Vitek 2 Compact system (bioMérieux, Marcy-l'Étoile, France). Isolates were stored in Skim milk-tryptone-glucose-glycerine stocks at -80° until further processing. Blood culture bottles, Vitek cards, and other microbiological reagents were funded and either locally procured or imported as part of the research study to ensure supply was uninterrupted during the study period.

### Bacterial isolates, sequencing and sequence analysis

All blood culture isolates from SPINZ that were identified as *Klebsiella* or *E. coli* were retrieved and revived in preparation for WGS. Frozen bacterial glycerol stocks were revived by streaking onto LB agar plates and incubating overnight at 37°C. A single colony pick was taken from each plate to inoculate LB broth and incubated overnight at 37°C. All procedures were performed under sterile conditions using aseptic techniques. DNA was extracted using the QIAamp DNA Mini Kit (QIAgen, Hilden, Germany), and shipped to the Quadram Institute of Bioscience, UK for sequencing. Libraries were

generated using a modified Nextera XT DNA protocol [24] and sequenced on an Illumina NextSeq 500 instrument (Illumina, California, USA).

Paired-end reads were de-novo assembled using Unicycler v0.5.0 [23] (depth filter: 0, mode: normal). Assemblies were filtered based on the KlebNET-GSP quality control criteria (i.e., contig count <500, genome size in the range 4,969,898–6,132,846 bp, and G + C content 56.35% to 57.98%). Assemblies meeting these criteria were uploaded to Pathogenwatch [25], and those not confirmed as *K. pneumoniae* species (n = 32), were excluded from further analysis. Where multiple *K. pneumoniae* blood culture isolates were available from the same neonate, only the first WGS-confirmed *K. pneumoniae* isolate recorded was included, thus excluding n = 2 assemblies.

Genome assemblies were further analysed using Kleborate v3.1.3 [26] to identify multi-locus sequence types (STs) and to determine the presence of AMR determinants. K and O loci (gene clusters encoding the capsule or lipopolysaccharide, respectively), and the associated K and O serotype predictions, were identified using Kaptive v3.1.0 [20]. Pathogenwatch was used to generate pairwise distances and a neighbour-joining tree of all *K. pneumoniae* genomes (based on 1,972 core genes [25]).

WGS data were matched to clinical metadata including sex, clinical outcome (discharged or death), date of NICU admission, date of specimen collection, and birth location.

For STs represented by at least four genomes, we undertook phylogenetic and transmission analysis. For each ST, the closest reference sequence to our isolates was identified using Bactinspector closest_match (v0.1.3) [27] (*K. pneumoniae* genomes updated from NCBI prior to running, June 2024). Accession numbers for all reference sequences are provided in S1 Table.

Snippy (v4.4.5) [28] was used to identify single nucleotide variants (SNVs) against the reference genome for each ST. The Coresnpfilter (v0.2.0) [29] package was used with a threshold of 80% (-c 0.8), to generate an alignment of all variant sites with an allele called in at least 80% of sequences. Invariant sites were included to create a final pseudo whole-genome alignment containing high-quality SNVs. Each alignment was filtered for recombination using Gubbins (v3.3.5) [30], and maximum-likelihood phylogenetic trees inferred from the resulting recombination-filtered alignment using RAxML-NG (v8.2.13) [31], with the GTR + G model and 1000 bootstrap repeats. The final bootstrapped tree was midpoint rooted using ggtree (v3.12.0) [32].

Transmission clusters were identified using the Transmission Estimator tool [33] taking as input a pairwise SNV distance matrix (generated for each ST from the recombination-filtered alignment, using snp-dists (v0.6.3) and pairwise temporal distances between infection dates (culture date where available, otherwise NICU admission date). Clustering via single-linkage with thresholds of 7–28 days and 5–25 SNVs, were tested. Only isolates with an available infection date were included in transmission/clustering analysis (148 isolates excluded).

In all genomes, contigs were investigated for plasmid indicators using the MOB-suite package (v3.1.9) [34]. Plasmids were classified into clusters, with clustered plasmids sharing the same generated ID across isolates. Novel plasmids were determined if the genomic distance deviated by >0.05 from a known reference. The tool Abricate (v1.0.1) [35] was used to detect the presence of AMR genes (using the CARD database [36]) on plasmids identified by MOB-suite.

To investigate *mph*A plasmids in ST307 clusters, we used the Bandage assembly graph viewer (v0.8) [37] to assess the plausibility of the nearest neighbour reference plasmid identified by MOB-suite being the location of the *mph*A gene. We used the blastn function in Bandage to search each graph for the reference plasmid sequence and the *mph*A gene, using the blastn parameter '-max_hsps 1' to report non-overlapping hits which we summed to calculate overall coverage and mean identity of the reference plasmid in each assembly.

## Statistical analysis

All statistical analyses were carried out in R (v4.4.0). The number of days to infection onset relative to admission was calculated for neonates with both a known date of NICU admission and culture date. This variable was categorised into 0

days through to 7 days, 8 + days and unknown, and into 'rapid-onset' (days 0–2) and 'delayed-onset' (days 3+) categories for comparison and ease of visualisation. Location of birth was categorized into 'inborn' if the neonate was born in any UTH site (including the UTH labour ward, UTH theatre and UTH postnatal wards), and 'outborn' for any other birth location. Study month was categorised into baseline and post-implementation, month 1–7 vs month 8–17, respectively.

Categorical variables with more than two groups were compared using the Kruskal–Wallis test. For binary variables, comparisons were conducted using the chi-squared test or Fisher's exact test when expected cell counts were small (<5). Differences in proportions (i.e., probabilities of success) across groups were assessed using the prop.test function from the *stats* package (v4.4.0). Univariate logistic regression was used to evaluate the association between birth location (inborn vs outborn) or onset category (rapid-onset vs delayed-onset) and cluster membership, using the glm() function (family = binomial) from the *stats* package. In all cases, a significant difference was considered if the p-value was < 0.05.

## Code and data availability

All analysis and visualisation code are available at https://github.com/klebgenomics/SPINZ (https://doi.org/10.5281/zenodo.17589084).

Whole-genome sequence data were deposited by the sequencing laboratory (Quadram Institute of Bioscience, UK) under BioProject PRJEB46513. Sample level accession information, together with associated clinical and source data, and WGS-derived typing data, is provided in S2 Table.

## Results

Of 465 blood culture isolates identified as *Klebsiella* species during the study period, 460 were successfully sequenced and 443 of these were confirmed from WGS as *K. pneumoniae*. Two isolates originally identified as *E. coli* were additionally characterised as *K. pneumoniae* from WGS data. Thus, we confirmed 445 blood-culture positive *K. pneumoniae* infections. Following quality control, a total of 411 confirmed high-quality *K. pneumoniae* genomes remained representing unique neonatal infections (S1 Fig). Twenty-four different *K. pneumoniae* STs were identified (Table 1).

The most prevalent was ST307 (69.3%, 285/411), followed by ST2004 (14.1%, 58/411). In total, there were nine common STs, each represented by ≥4 infection isolates. There was a close association between K locus and ST, with each ST associated with a single K locus. KL102 was the most commonly identified K locus, exclusively found in all ST307 isolates (Table 1). The second most prevalent was KL23, found in all ST2004 isolates. The most prevalent O type was O2, representing 72.3% (297/411) of isolates (Table 1). Each ST was associated with a single O type, except for ST307 in which 19.6% (n = 56/285) of isolates were O2α and the remainder O2β (n = 229/285).

The presence and number of acquired AMR genes varied between and within STs (Fig 1). Few isolates had an acquired carbapenemase gene (n = 11/411, 2.7%; all ST5856 (n = 9) or ST340 (n = 2) carrying $bla_{NDM-5}$). Most isolates had an acquired extended-spectrum ß-lactamase (ESBL) gene (99.0%, 407/411; n = 405 with only $bla_{CTX-M-15}$ in 22 STs, n = 1 ST5856 with both $bla_{CTX-M-15}$ and $bla_{TEM-116}$, n = 1 ST101 with $bla_{CTX-M-14}$). Many isolates had additional acquired AMR genes detected, including those associated with resistance to ß-lactams ($bla_{OXA-1}$ and $bla_{TEM-1}$, 99.3%; supplementing the intrinsic core gene $bla_{SHV}$), fluoroquinolones (81.5%), aminoglycosides (100%), tetracycline (75.7%), phenicols (12.9%), macrolides (10.7%), sulfonamides (99.5%) or trimethoprim (94.9%). No acquired AMR determinants for colistin, fosfomycin, or tigecycline were detected. The virulence-associated yersiniabactin siderophore locus (*ybt*) was detected in 10.5% of isolates (n = 43/412). We did not identify any isolates with hypervirulence-associated loci aerobactin, salmochelin, colibactin, *rmpADC* or *rmpA2*.

Of the 411 *K. pneumoniae* isolates successfully sequenced and representing unique culture-positive neonates, 324 had associated clinical metadata available which included the date of NICU admission, and 263 of these had a specimen collection date recorded for the blood sample from which *K. pneumoniae* was cultured (Table 2). For neonates with both dates, the median day of onset of confirmed *K. pneumoniae* infection relative to admission was three days for both the baseline and

**Table 1. K and O loci and types identified per sequence type (ST).**

| ST | K locus | K type | O locus | O type | Count |
|---|---|---|---|---|---|
| ST307 (n=285) | KL102 | Capsule null (n=249) | OL2α.2 | O2β | 193 |
| | | | OL2α.2 | O2α | 56 |
| | | Unknown (n=36) | OL2α.2 | O2β | 36 |
| ST2004 | KL23 | K23 | OL2α.2 | O2β | 58 |
| ST983 | KL127 | Unknown | OL13 | O13 | 12 |
| ST5856 | KL15 | K15 | OL4 | O4 | 9 |
| ST101 | KL17 | K17 | OL2α.1 | O1αβ,2α | 9 |
| ST985 | KL39 | K39 | OL2α.2 | O1αβ,2β | 6 |
| ST15 | KL54 | K54 | OL2α.2 | O1αβ,2β | 4 |
| ST147 | KL64 | K64 | OL2α.1 | O2α | 4 |
| ST1119 | KL110 | Unknown | OL3γ | O3γ | 4 |
| ST1486 | KL38 | K38 | OL3γ | O3γ | 3 |
| ST340 | KL15 | K15 | OL4 | O4 | 2 |
| ST268 | KL20 | K20 | OL2α.1 | O2α | 2 |
| ST45 | KL24 | K24 | OL2α.1 | O2α | 2 |
| ST14 | KL2 | K2 | OL2α.1 | O1αβ,2α | 1 |
| ST1427 | KL67 | K67 | OL13 | O13 | 1 |
| ST13 | KL3 | K3 | OL2α.2 | O1αβ,2β | 1 |
| ST1731 | KL122 | Unknown | OL2α.2 | O2β | 1 |
| ST20 | KL24 | K24 | OL2α.1 | O1αβ,2α | 1 |
| ST253 | KL64 | K64 | OL2α.1 | O1αβ,2α | 1 |
| ST280 | KL23 | K23 | OL2α.2 | O2β | 1 |
| ST405 | KL151 | Unknown | OL4 | O4 | 1 |
| ST54 | KL14 | K14 | OL3γ | O3γ | 1 |
| ST831 | KL18 | K18 | OL2α.1 | O1αβ,2α | 1 |
| ST966 | KL71 | K71 | OL3γ | O3γ | 1 |

post-implementation periods, and the proportion of sepsis classified as rapid-onset was similar in both periods (43.9% baseline vs 36.8% post-implementation, p=0.36). Amongst the 324 *K. pneumoniae* culture-positive neonates with clinical data, 279 had a recorded location of birth, with 205 inborn at UTH and 74 outborn (Table 2). The proportion of culture-positive neonates inborn at UTH was significantly lower in the post-implementation period (64.7% post-implementation vs 77.3% baseline, p=0.04).

Of the 324 sequenced *K. pneumoniae* isolates with clinical metadata, the majority (n=228, 70%) were isolated in the baseline period (based on date of specimen collection where available or NICU admission date otherwise) (see Table 2, Fig 2). ST307 *K. pneumoniae* were present from the beginning of the study, with a median 5 infections per week throughout the baseline period (mean 6.7 infections per week for 31 weeks). The number of sequenced infections with ST307 declined to zero after the IPC was implemented (mean 0.6, median 0 infections per week, weeks 32–73) (Fig 2). However, ST307 infections reappeared in weeks 54 and 56 (n=3) and weeks 72 and 73 (n=6). A small number of sequenced infections with the second most common sequence type, ST2004, occurred prior to the implementation period (n=8, mean 1.3, median 1 infection per week over 6 weeks). However most sequenced infections caused by ST2004 occurred during the post-implementation period (n=39, mean of 2.4 and median of 2 infections per week over a 16-week period) (Fig 2). ST101 was also associated with infections during both the baseline (n=4) and post-implementation periods (n=5). In contrast, ST985 (n=5) and ST147 (n=4) were identified amongst sequenced isolates from the baseline period only, and ST983 (n=11) and

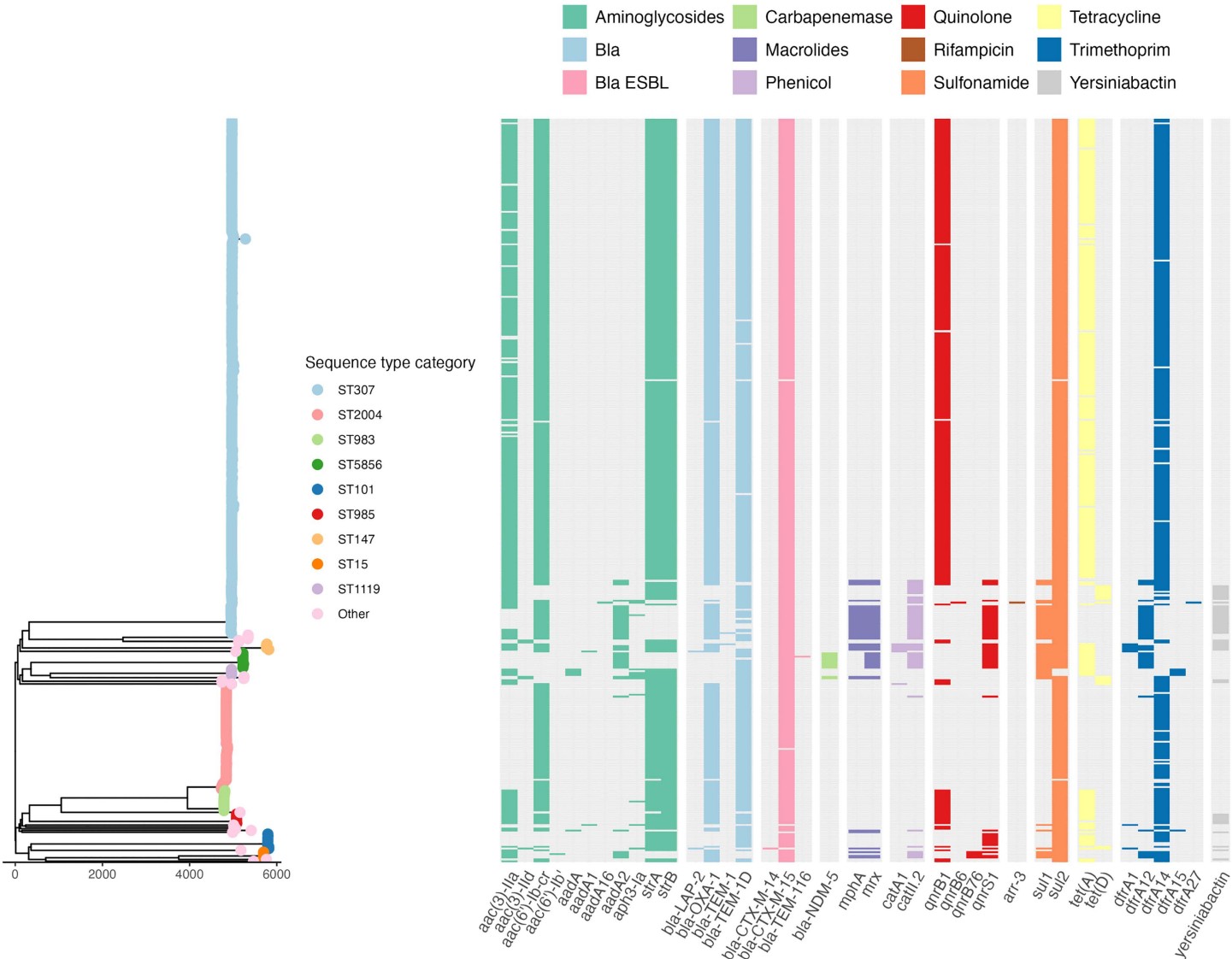

**Fig 1. Neighbour-joining tree of WGS-confirmed *K. pneumoniae*.** ST is indicated by the colour of tree branch tips. 'Other STs' (pink tips) are those with ≤4 genomes identified: ST1486, ST340, ST268, ST45, ST14, ST1427, ST13, ST1731, ST20, ST253, ST280, ST405, ST54, ST831 and ST966. Scale bar indicates genetic distance (SNVs). Heatmap indicates the presence (coloured block) or absence of acquired resistance genes corresponding to specific antibiotic classes (indicated by the block colour). Presence of the virulence-associated yersiniabactin locus is shown in the final column (grey).

ST15 (n = 4) were identified post-implementation only (Fig 2). ST983 infections were temporally clustered within a short time period post-implementation (n = 11 between weeks 33 and 38, mean 1.8 and median 1.5 per week), and ST15 was detected post-implementation only, in weeks 71–73 (n = 4, 1–2 per week for 3 weeks) (Fig 2). Eight other STs were detected during the study period, mostly as single infections occurring sporadically during the study (n = 10, see Fig 2).

## Transmission clusters

As the timeline of sequenced cases revealed temporal clustering by ST (Fig 2A), we undertook high-resolution genomic comparisons to investigate potential transmission patterns and explore changes following the IPC intervention (see

**Table 2. Clinical characteristics of neonates with invasive *Klebsiella pneumoniae* disease during the baseline and post-implementation study periods.**

| Characteristic | N[1] | Baseline N = 228[2] | Post-implementation N = 96[2] |
|---|---|---|---|
| **Sex** | 279 | | |
| Male | | 94 (48%) | 51 (60%) |
| Female | | 100 (52%) | 34 (40%) |
| Unknown | | 34 | 11 |
| **Admission Date** | 324 | | |
| Present | | 228 (100%) | 96 (100%) |
| **Culture Date** | 263 | | |
| Present | | 187 (100%) | 76 (100%) |
| Unknown | | 41 | 20 |
| **Clinical outcome** | 273 | | |
| Discharged | | 121 (64%) | 45 (54%) |
| Died | | 68 (36%) | 39 (46%) |
| Unknown | | 39 | 12 |
| **Birth location** | 279 | | |
| Inborn | | 150 (77%) | 55 (65%) |
| Outborn | | 44 (23%) | 30 (35%) |
| Unknown | | 34 | 11 |
| **Days to onset from admission** | 263 | | |
| 0 | | 16 (8.6%) | 3 (3.9%) |
| 1 | | 33 (18%) | 12 (16%) |
| 2 | | 33 (18%) | 13 (17%) |
| 3 | | 35 (19%) | 12 (16%) |
| 4 | | 20 (11%) | 9 (12%) |
| 5 | | 15 (8.0%) | 3 (3.9%) |
| 6 | | 11 (5.9%) | 2 (2.6%) |
| 7 | | 12 (6.4%) | 1 (1.3%) |
| 8 + days | | 12 (6.4%) | 21 (28%) |
| Unknown | | 41 | 20 |

[1] N = Total with non-missing data for this variable. [2] N (%) = Total with this value (values in brackets indicate percentage of total samples with non-missing data for this variable, N).

Methods). Single-linkage clustering was done using a genetic distance threshold of ≤10 core genome SNVs and a temporal distance threshold of ≤28 days. Singleton STs were designated as non-clustered cases. Using these thresholds, clusters ranged in size from 2–202 neonates (median 6, IQR 4–11), and duration from within a single week to 33 weeks (median 3, IQR 2–5).

Overall, the proportion of cases belonging to clusters, or attributable to clusters, was significantly higher in the baseline period than the post-implementation period (n = 221/226, 97.8% vs n = 86/95, 90.5%, p=<0.01 and n = 214/226, 94.7% vs n = 78/95, 82.1% attributable, p=<0.001). To assess sensitivity of clustering to choice of thresholds, we additionally ran clustering with a range of pairwise distance thresholds, between 5–25 genome-wide SNVs and between 7–28 days, which yielded only minor differences in cluster number and size (S1 Appendix). Varying SNV distance threshold had no impact on clustering for ST307, ST2004, ST983, ST147, ST15 and ST985. Only for ST101 did a SNV distance threshold of five have a minor impact on clustering compared to 10–25 SNVs. Varying the temporal threshold between 7–28 days had a

A

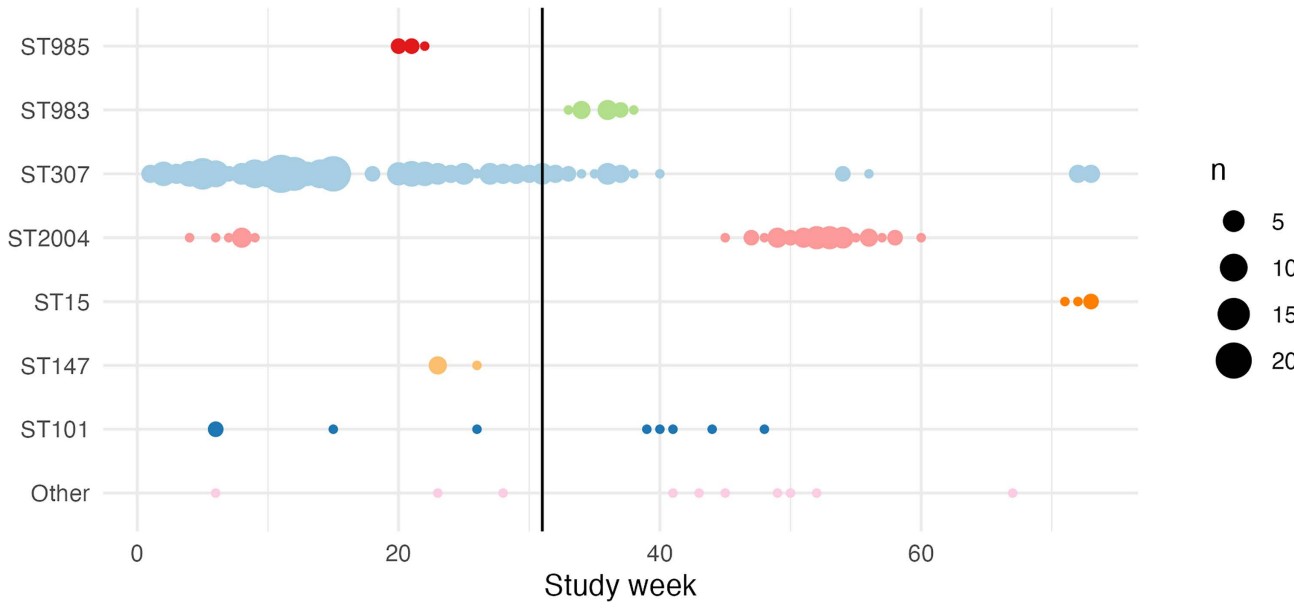

B

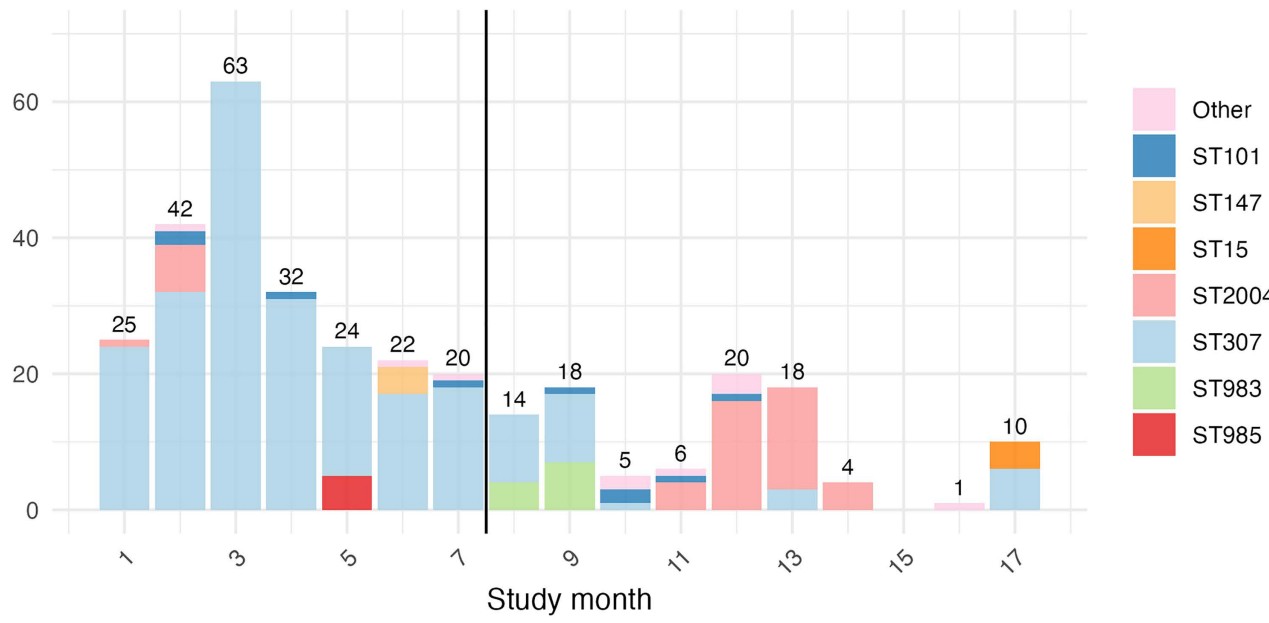

**Fig 2. Temporal distribution of WGS-confirmed *Klebsiella pneumoniae* invasive infections.** Solid vertical lines represent the start of the post-implementation period, which continues to the study end. (A) Timeline shows number of sequenced *K. pneumoniae* cases per week, belonging to each ST (one row per ST, also coloured by ST to match panel B). Each data point indicates one week, sized to indicate the number of isolates of the given ST per week. (B) Monthly counts of confirmed *K. pneumoniae* cases, stratified by ST. Total counts of cases per month are represented on each bar.

small impact on the number of clusters and proportion of clustered sequences for ST307 ranging from (97.0% (225/232) in 8 clusters - 99.5% (231/232) in 5 clusters), ST101 (22% (2/9) in 1 cluster - 67% (6/9) in 2 clusters), ST2004 (94% (44/47) in 6 clusters - 100% (47/47) in 2 clusters), ST983 (100% (9/9) in 2 clusters - 100% (9/9) in 1 cluster) and ST147 (75% (3/4) in 1 cluster - 100% (4/4) in 1 cluster). In contrast, varying the temporal threshold had no impact on clustering for ST15 and ST985. Based on the range of estimates from these sensitivity analyses, between 292–306 (94.2–98.7%) of total cases were associated with a transmission cluster, and 272–293 out of 310 (87.7–94.5%) were attributable to nosocomial transmission.

Clustered sequences were not significantly associated with inborn vs outborn neonates (OR 1.23, 95% CI 0.32-3.89, p = 0.7), or rapid-onset infection (OR 1.47, 95% CI 0.44-5.63, p = 0.5). Of the 202 inborn neonates within the study hospital with a *K. pneumoniae* infection, 193 (95.5%) were within a cluster, and 9 (4.5%) were not. Proportions were similar in outborn neonates in which 70/74 (94.6%) of infections were clustered. Of the 110 neonates with rapid-onset infection, 106 (96.4%) isolates were clustered, and similarly, 144/152 (94.7%) isolates from neonates with delayed-onset infection were clustered. For neonates with data in both (n = 236), inclusion of an interaction term between birth location and onset category did not yield a statistically significant effect (p = 0.3), despite changes in the estimated odds ratios.

During the baseline period, all identified clusters included infections in both inborn and outborn neonates (Fig 3A). ST147 cluster 1 comprised exclusively delayed-onset infections (n = 4; 3 inborn, 1 outborn), and ST985 cluster 1 was predominantly delayed-onset (4/5; 2 inborn, 2 outborn), with one rapid-onset case in an outborn neonate. Neither ST was observed in post-implementation clusters. Metadata for ST2004 cluster 1 was limited, 2 of 8 neonates were outborn with delayed-onset infections; one additional case involved a neonate with delayed-onset infection and unknown birth location, and another was an inborn neonate with an unknown onset category infection (Fig 3A). In contrast, post implementation ST2004 cluster 2 infections included both rapid-onset (8/39) as well as delayed-onset (22/39) infections, occurring predominantly in inborn (21/39) compared to outborn (16/39) neonates (Fig 3B).

ST307 cluster 1 was the largest, comprising 202 infections, predominantly among inborn neonates (68.8%, 139/202). Among those inborn neonates, rapid- and delayed-onset infections occurred in equal proportions (42.4%, 59/139 for both) similarly to proportions in outborn neonates (35.3% rapid-onset, 52.9% delayed-onset). ST307 cluster 2 involved equal numbers of inborn and outborn (n = 3 each) neonates, with most infections being delayed-onset (5/7; 3 inborn, 1 outborn, 1 unknown). ST307 cluster 3, first detected at baseline (n = 1), persisted into the post-implementation period and included both inborn (n = 5) and outborn (n = 4) neonates, with 50% each of rapid-onset and late-onset infections (Fig 3A).

Two novel ST307 clusters emerged post-implementation (Fig 3B). Cluster 4 caused exclusively delayed onset infections (n = 3), affecting one inborn and two outborn neonates—distinct from the mixed onset pattern seen in earlier ST307 clusters. In contrast, cluster 5 predominantly caused rapid-onset infections (4/6), all in inborn neonates. Additional post-implementation clusters included ST101 cluster 2, which caused delayed-onset infections in two inborn neonates. Two new clustered STs, ST15 cluster 1 which included rapid-onset infections in inborn neonates (n = 2) and delayed-onset infections in outborn neonates (n = 2). The second, ST983 cluster 1, was predominantly detected in inborn neonates (10/11), causing both rapid- (n = 6) and delayed-onset (n = 2) infections (Fig 3B).

## Persistence vs novel introductions

Three STs (ST307, ST2004 and ST101) were identified both at baseline and post-implementation. To explore whether post-implementation infections may have arisen from contamination persisting during the intervention or represented new strain introduction into the unit, we conducted additional phylogenomic analyses (see Methods).

The ST307 phylogeny revealed a dominant sublineage accounting for n = 248/285 ST307 genomes (cluster 1, green in Fig 4). Of these, 202 had known dates of admission and 180 had a known specimen collection date; these all belonged to cluster 1 and were detected throughout the baseline period (n = 31 weeks, median 5 cases per week) and during the first three weeks post-implementation, but not after month eight. Four other ST307 sublineages were evident in the tree,

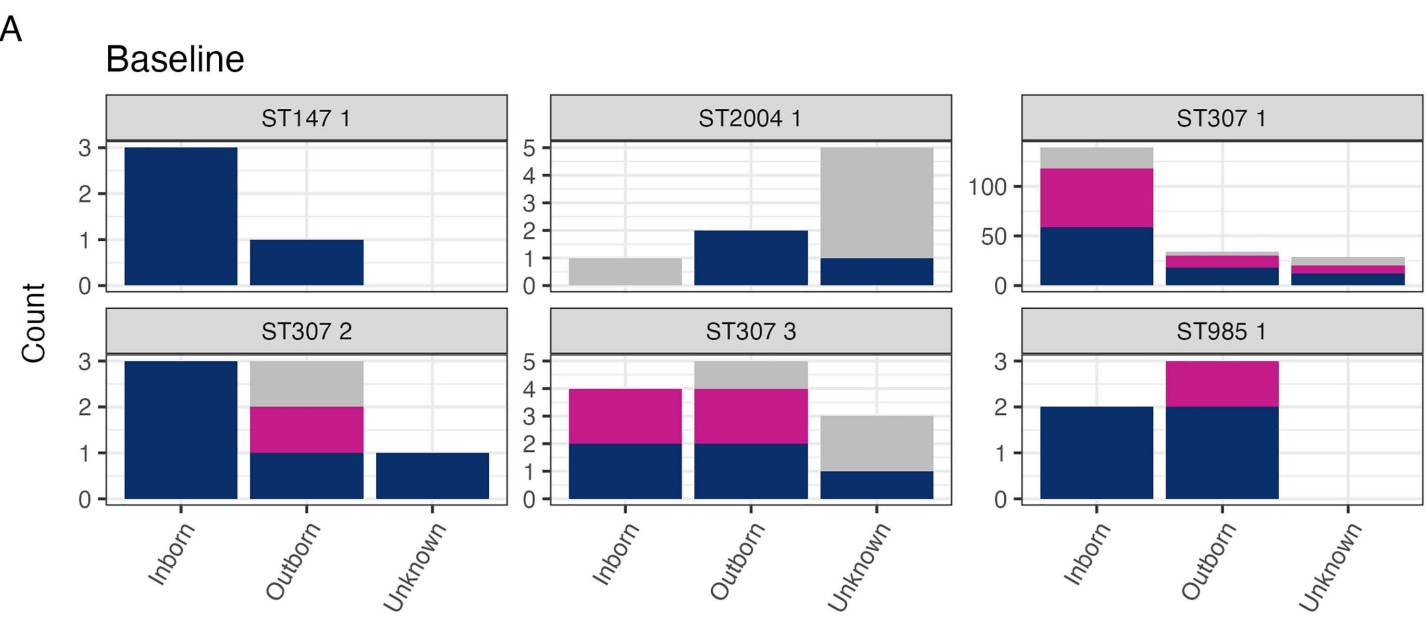

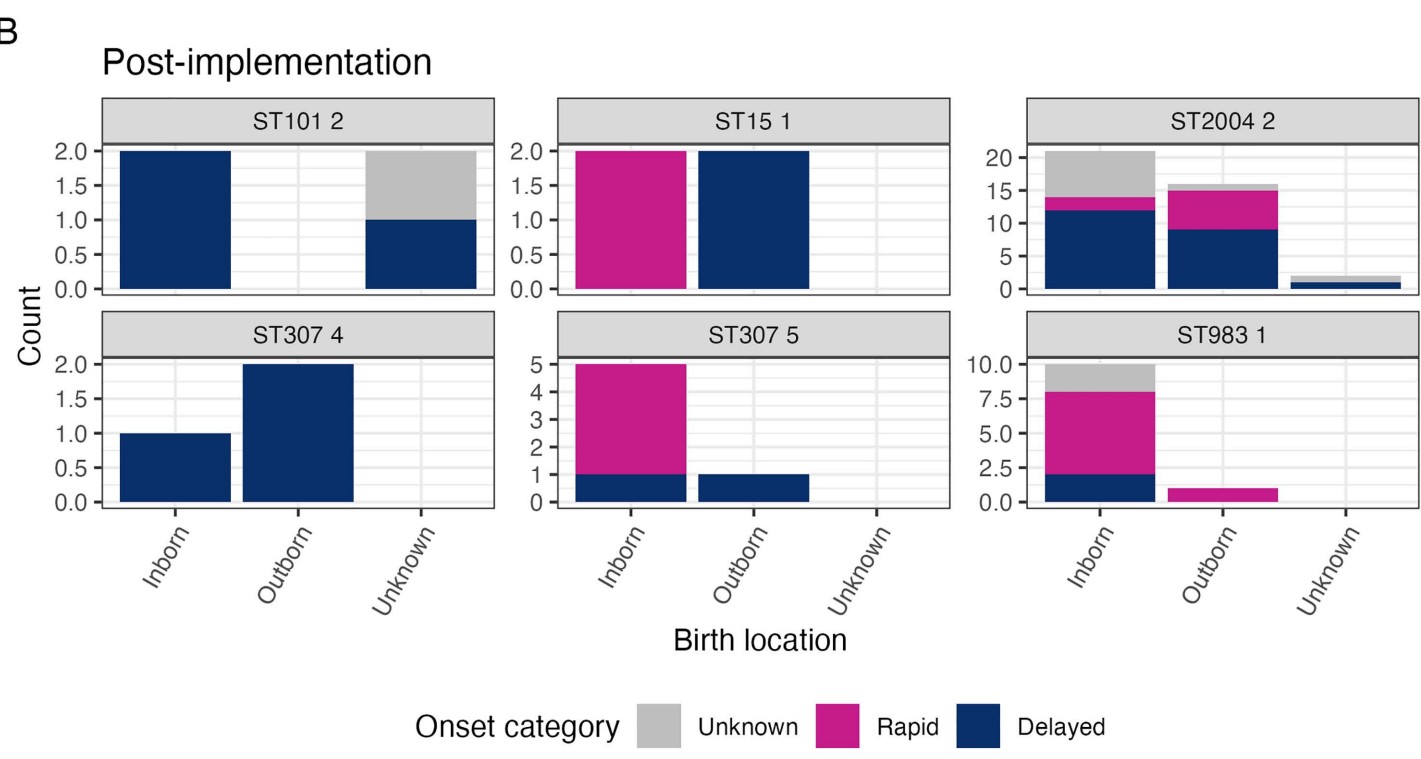

**Fig 3. Breakdown of birth location and onset category for infections in each cluster.** Infections are divided into those detected **(A)** at baseline and **(B)** post-implementation. Individual plots are labelled with ST and cluster number. Clusters were assigned to baseline or post-implementation based on the timing of the first case in the cluster. Onset category is defined as 'rapid-onset' (day 0-2 of admission to NICU) or 'delayed-onset' (day 3+ of admission to NICU). ST101 cluster 1 is not shown due to lack of any data on birth location or onset category for all cases.

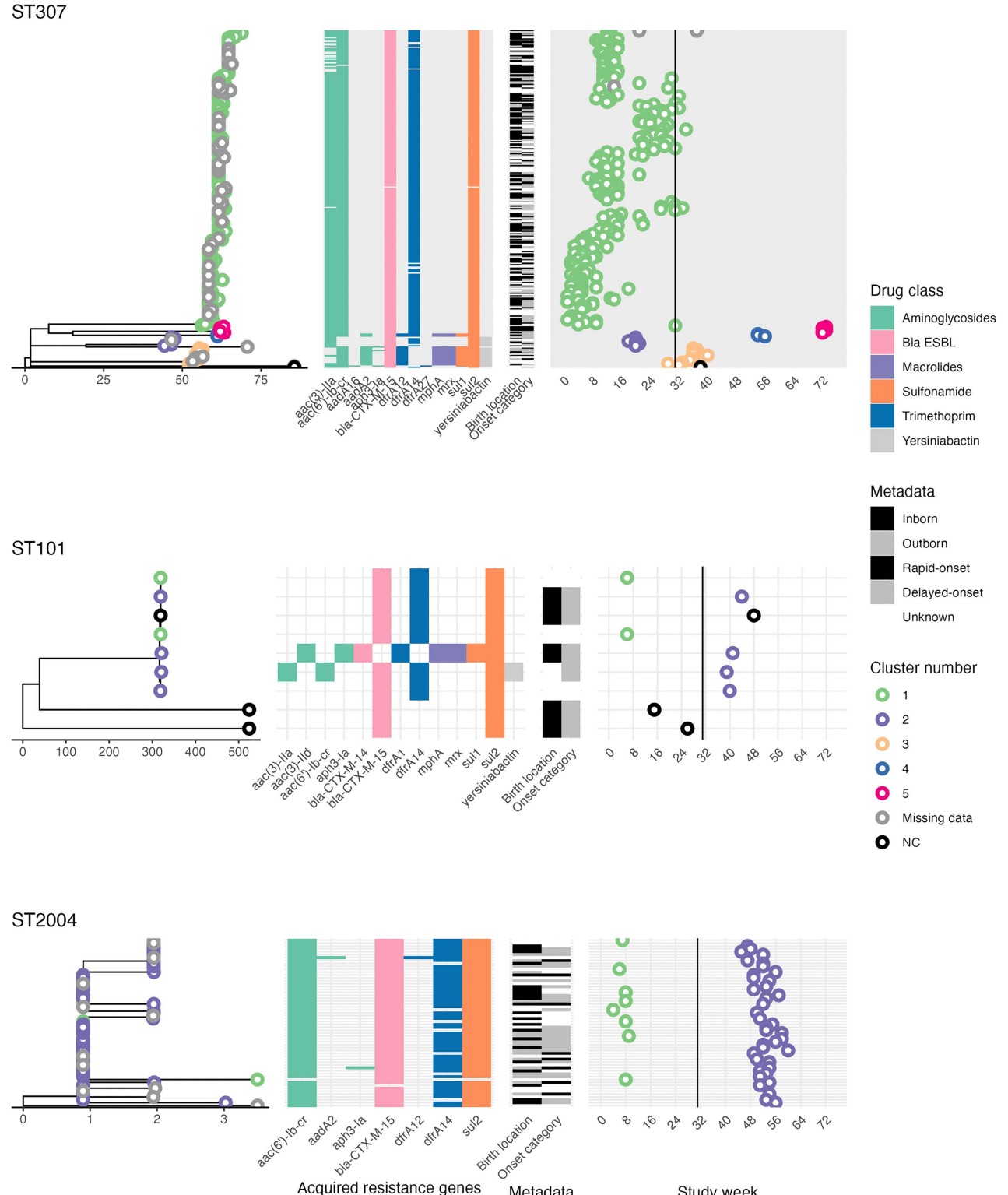

**Fig 4. Core SNV phylogenies of *Klebsiella pneumoniae* aligned to acquired AMR gene data and infection timelines. ST307** isolates (top), **ST101** isolates (middle) and **ST2004** isolates (bottom). A midpoint-rooted maximum-likelihood phylogenetic tree was inferred separately for each ST, based on an alignment of SNVs identified against a closed reference genome of that ST. Tips are coloured to indicate the cluster number, inferred via

single-linkage clustering using thresholds of ≤10 SNVs and ≤28 days. Scale bar indicates genetic distance (SNVs). Heatmaps show the presence of key acquired resistance genes and hypervirulence genes (if present) aligned to tree tips. Timeline plot indicates the date of isolation (study week) for each isolate, aligned to tree tips and coloured by cluster number. The timing of the start of the IPC implementation is indicated by a solid vertical line.

mapping to clusters 2–5 (Fig 4). Cluster 3 was first detected two weeks prior to the IPC intervention (n = 1) and persisted into the post-implementation period (n = 11, median 1 per week for 9 weeks) but was not detected after this. One other cluster (cluster 2) was detected prior to intervention (n = 7, median 1 per week for four weeks); and two clusters were detected during the post-implementation period (cluster 4; n = 3, median 1.5 per week for three weeks in month 13; and cluster 5; n = 6, median three per week for two weeks in month 17) (Fig 4). No ST307 cases sequenced from the post-implementation period (cluster 4 and 5) belonged to the same phylogenetic lineage as baseline isolates (cluster 1, 2 and 3), and the minimum pairwise genetic distance between cluster 4 or 5 and any other isolates was 71 SNVs. Therefore, the ST307 infections identified post-implementation appear to be unrelated to the ST307 sublineage responsible for the earlier infection clusters.

To assess the impact of using admission date as a proxy for date of *K. pneumoniae* isolation (in n = 61 cases with an admission date, but lacking a recorded culture date), an additional clustering sensitivity analysis was conducted for ST307 (S2 Fig). Clustering based solely on isolates with a known culture date (S2B Fig) identified the same isolates present in clusters, but separated the very large cluster into two, separated by an intervening period of 32 days between culture dates in month 5 and month 6.

All except one ST307 isolate carried $bla_{CTX-M-15,}$ and the different subgroups of ST307 presented overall similar AMR profiles, with the exception of cluster 4 in which n = 9/12 isolates carried *aadA2* instead of *aac (3')-IIa* (suggesting loss of gentamicin resistance) and *dfrA12* instead of *dfrA14* (retention of trimethoprim resistance). Cluster 3 and 4 additionally carried *mph*A and *mrx* genes (suggesting reduction of azithromycin susceptibility), and *sul1* in addition to *sul2* (retention of trimethoprim-sulfamethoxazole resistance). The acquired siderophore, yersiniabactin, was present in clusters 2 and 4 only (Fig 4).

The ST101 core SNV phylogeny revealed a sub-lineage of seven genetically related isolates, and two genetically distant isolates (Fig 4). The genetically-linked isolates were temporally divided into two clusters, one during the baseline period (n = 2 cases, isolated in the same week) and a second during the post-implementation period (n = 4 cases over 6 weeks) (Fig 4). The seventh isolate was just seven SNVs different and detected four weeks after the last cluster-2 case. Clusters 1 and 2 were not genetically distinct, separated by 0–7 pairwise SNVs and sharing the same AMR profile, consistent with persistence of a single clone from the baseline to the post-implementation period (Fig 4).

The ST2004 phylogeny revealed a dominant sub-lineage, split into two temporally separated clusters of genetically near-identical isolates (Fig 4). Cluster 1 was detected during the baseline period (n = 8 cases, median 1 per week for 6 weeks), and cluster 2 in the post-implementation period (n = 39 cases, median 2 cases per week for 16 weeks) (Fig 4). Pairwise genetic distances between these two clusters were 0–5 SNVs, with identical resistance profiles, again consistent with long-term persistence of a single clone.

The remaining four STs with ≥4 infection isolates (ST15, ST147, ST983 and ST985), detected in either the baseline period or the post-implementation period, could not be phylogenetically represented due to low intra-sequence diversity. For each of these STs, the maximum number of SNVs between all genomes was three (full SNV data is available at https://github.com/klebgenomics/SPINZ). AMR profiles and infections by study week for these STs are shown in S3 Fig.

### Serotypes and AMR post intervention

The prevalence of predicted K-types differed significantly between the baseline and post-implementation periods (p < 0.001 using Chi-square test) (Fig 5A), driven by changes in the prevalence of associated STs. Most notably, infections

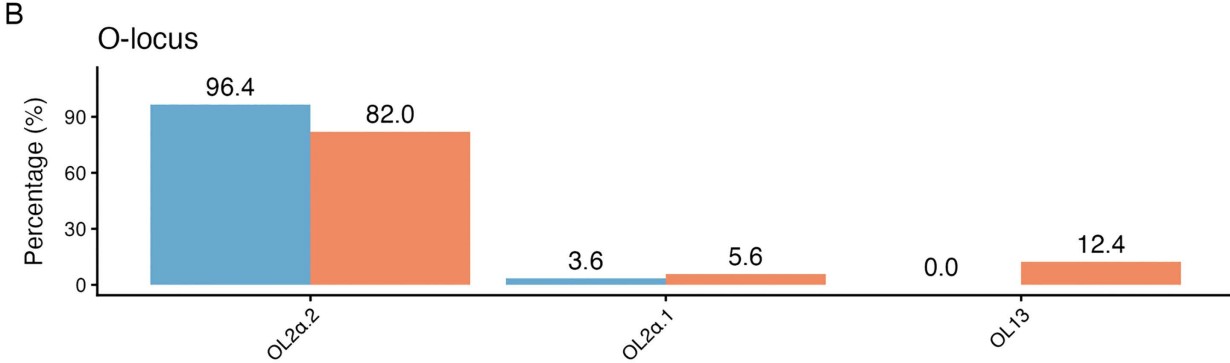

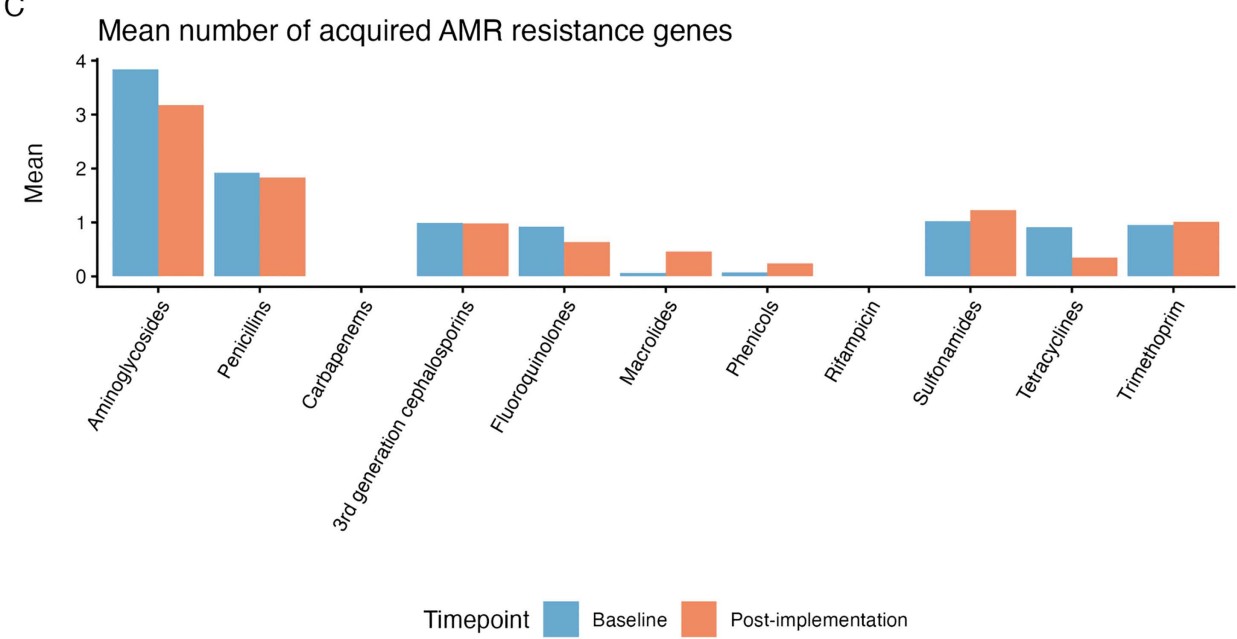

**Fig 5. Prevalence of K-loci (A), O-loci (B), and mean number of acquired AMR genes (C) amongst *K. pneumoniae*.** Infections split into baseline and post-implementation study timepoints.

in the baseline period were dominated by a large cluster of ST307 with disruptions in the KL102 locus, which were assigned by Kaptive as capsule null (87.1% of infections), whereas the post-implementation period was dominated by KL23 ST2004 (43.8%), KL102 ST307 (24.7%) and KL127 ST983 (12.4%). The KL102-disrupted ST307 isolates belonging to ST307 cluster 1, shared a deletion of 661 base pairs truncating the *wbaP* and *wzy* genes within the capsule locus (S4 Fig) that are likely to result in a lack of capsule synthesised via this locus. The distribution of predicted O loci also differed in the post-implementation periods compared with baseline (Fig 5B), most notably with reduced prevalence of OL2α.2 loci (96.4% vs 82.0%) and introduction of O13 (0% vs 12.4%) via the ST983 cluster.

AMR gene profiles were very similar across STs and clusters (Fig 1), and consequently there were few changes post-implementation (Fig 5C). There was an increase in presence of azithromycin resistance-associated genes post-implementation (from 3.1%, n = 7/228 to 22.9%, n = 22/96), driven by the presence of *mphA* in the post-implementation ST307 clusters 3 and 4. In ST307 cluster 4 genomes, the *mphA* gene was carried on a 6 kilobase pairs (kbp) contig flanked by repeat sequences (transposases). The location could therefore not be fully resolved from short read sequence data, however inspection of the assembly graph suggests it was likely located in a plasmid with similarity to a *mphA*-carrying reference plasmid with GenBank accession CP021752.1 (identified as a close reference using MOB-suite), as this plasmid was present in the assembly graphs (96.9% coverage at mean 98.6% identity) and connected the *mphA* contig (S5 Fig). The location of *mphA* was even harder to resolve in cluster 3 genomes, where it was located on a 3.2 kbp contig. However, the reference plasmid CP021752.1 had only 74% coverage in these genomes, suggesting this cluster may harbour a distinct *mphA* plasmid.

## Discussion

Analysis of high quality WGS data from 411 *K. pneumoniae* blood isolates from neonates showed that most infections (95.6%) belonged to temporal clusters of near-identical isolates, consistent with nosocomial transmission. The fraction of clustered isolates was the same amongst neonates inborn at UTH and those born elsewhere before admission to the NICU, with clustering not shown to be associated with birth location, rapid-onset of infection (day 0–2 of NICU admission), or their interaction. This finding is consistent with nosocomial transmission in the NICU specifically (as opposed to, e.g., the labour ward).

The primary SPINZ analysis reported previously that sepsis incidence declined following implementation of the IPC bundle [22]. Here, the new *K. pneumoniae* WGS data support that the IPC bundle successfully disrupted a large, long-running outbreak of *K. pneumoniae* ST307 that persisted throughout the baseline observation period (median 5 new infections per week) but disappeared three weeks after the implementation began. The data also showed the fraction of *K. pneumoniae* infections involved in clusters to be smaller post-implementation (97.8% vs 90.5%, p=<0.01), consistent with an effect of the implementation. However, the overall cluster rate remained high compared with similar estimates from other African NICUs [38] and indicated ongoing challenges with maintaining effective IPC.

Notably, a substantial proportion of infections (26.8%), and of clustered infections (34.5%), were rapid-onset, identified within two days of admission to the NICU. This differs from the usual definition of healthcare-associated infection (HCAI) which designates infections occurring after a hospital stay of two days or more [39]. However, here we observed multiple cases of neonates with *K. pneumoniae* isolated from blood stream infections on day 0 or 1 of NICU admission, with isolates genetically identical to those from other patients on the same ward within days.

This includes outborn infants, providing compelling evidence of transmission within the NICU as opposed to acquisition from a common source outside the unit prior to admission. Recent outbreak investigations in other African NICUs have identified diverse sources of infection including contaminated environments (e.g., sinks, bed railings), water sources and colonised healthcare workers as well as contaminated reagents including intravenous (IV) bags [40–42]. It is plausible that outbreaks mediated by contaminated reagents, which are directly inoculated into newborns via injection or feeding tubes, could result in rapid-onset infections. In contrast, transmission mediated by environmental contamination of colonised

healthcare workers may be more likely to begin with gut colonisation of neonates, followed by overgrowth in the gut before progressing to systemic infection and sepsis, better fitting the standard expectations of HCAI onset after ≥2 days on the ward. Consistent with this, the distribution of rapid-onset infections was skewed, with six putative transmission clusters including very few (0 or 1) rapid-onset infections, while four clusters—most notably the large, persistent ST307 baseline cluster 1—were dominated by rapid-onset infections, often including babies born outside the hospital. During the baseline period, most clusters were composed of delayed-onset infections (≥2 days post admission). However, following implementation of the IPC bundle, 3 out of 6 clusters included at least half, or a majority, rapid-onset infections. This shift may indicate the IPC measures effectively eliminated outbreak sources linked to environmental contamination or poor hand hygiene, while more direct sources, such as contaminated reagents, may have persisted. This explanation would be consistent with the nature of the IPC bundle, however no environmental or reagent screening was undertaken during the study period to enable a direct assessment of outbreak sources post-implementation, and whether they differed in nature to the baseline period sources. This raises an important consideration for future studies assessing the effectiveness of IPC interventions.

Infections with ST101 and ST2004 were associated with clusters in both the baseline and post-implementation periods, with genetically indistinguishable strains isolated in both periods. For ST101, we observed two infections in the same week (week 6 during baseline), then four infections during weeks 39–44 in the post-implementation period (0–7 SNVs from the first two cases, see Fig 4B and S6 Fig), and a single infection 4 weeks later (7 SNVs). The timing of onset information was unknown for the two baseline infections, however the second cluster of infections were all delayed onset (day 5–15 following NICU admission) and the single infection in week 48 was also delayed onset, following 41 days in the NICU (i.e., the infant was already present on the ward during the 4-case cluster, so could have become colonised around the same time as the others). For ST2004 we observed a cluster of eight infections in weeks 4–9 and a second cluster of 39 infections between weeks 45–60, with 0–5 SNVs between the two clusters. For the first cluster, timing of onset was known for three babies, all of whom had delayed-onset (3–7 days following admission to NICU); the later cluster included eight rapid-onset infections (in two inborn babies and six outborn) and 22 that were known to be delayed onset (3–39 days). We hypothesise these patterns could be explained by some form of environmental contamination present during the baseline period, which persisted through the IPC intervention, and occasionally transmitted to additional sources that resulted in temporally clustered infections.

The high numbers of cases of neonatal sepsis with *K. pneumoniae* overall and diversity of STs (24 among 411 isolates) over a 19-month period is similar to findings in other settings [43]. The high level of diversity can be linked to the various reservoirs that could be introducing pathogens such as *K. pneumoniae* into the NICU environment, including healthcare workers, parents, medical equipment, and other environmental fomites [44]. Some of the *K. pneumoniae* lineages associated here with nosocomial transmission (ST307, ST101, ST147, ST15) are considered high-risk clones that have been associated with HCAI outbreaks in other settings including neonatal units across continents as well as within adult ICUs in high-income countries [45,46]. In particular the most common ST, associated with five clusters and a majority of infections, was ST307. Since its emergence in the mid-1990s [47], ST307 has rapidly spread across every continent except Antarctica, causing nosocomial outbreaks worldwide [48–50]. One such outbreak in a Korean NICU was reportedly controlled through enhanced IPC measures, including frequent disinfection of medical devices, active surveillance cultures, hand hygiene re-education, and segregation of infected and newly admitted individuals [51]. Although *Klebsiella pneumoniae* ST307 has been reported in Africa, including Zambia, it appears less frequently than in Europe and the USA. To our knowledge this is the first study to perform genomic characterization using WGS, including sequence typing and AMR gene profiling, to explore the prevalence and distribution of STs in Zambia.

Notably, the ST307 strain causing the very large cluster (>200 infections during baseline observation) had a disrupted KL102 locus (S4 Fig), which was predicted as capsule-null by the Kaptive genotyping tool [20].The two truncated genes, *wbaP* and *wzy*, are thought to encode proteins essential for capsule formation, responsible for the initiation of

the synthesis of the capsular repeat units and their polymerisation, respectively [52]. Capsule-null *K. pneumoniae* are rarely observed in blood isolates, consistent with the role of the capsule in evading serum complement and phagocytosis [53,54]. However capsule inactivation has been reported in clinical isolates from other body sites, and may enhance epithelial cell invasion, biofilm formation and persistence particularly in urinary tract infections [55]. ST307 genomes harbour a second putative polysaccharide locus, Cp2 [56], that was intact in all cluster 1 ST307 genomes, and may encode an alternative capsule. We were unable to retrieve isolates to experimentally confirm whether cluster 1 ST307 isolates were encapsulated. Previous reports of a KL102-disrupted ST307 with intact Cp2 loci display reduced complement resistance when tested against sera from healthy adult volunteers (4-log reduction after two hours). However, this reduced level of resistance may still be sufficient to infect neonates, whose complement systems are underdeveloped [57]. Given the susceptibility of unencapsulated bacteria to host immune defences, it seems unlikely these isolates lacked a capsule yet maintained sustained transmission at the observed scale over a sustained period of time, although it is conceivable if the bacteria were directly inoculated into the bloodstream via contaminated IV fluids, as has been reported in other LMIC NICU settings [42]. This observation, alongside the dynamic shifts in K-type prevalence, underscores the importance of considering both O- and K-antigen variability in the design of a maternal vaccine, and also the relevance of antibody-based immunity to bacterial strains directly introduced to the bloodstream via contaminated reagents or equipment. The detection of a second capsule locus in ST307—and potentially in other sequence types—raises concerns about selective pressure driving capsule switching or alternative polysaccharide expression, potentially undermining vaccine efficacy. Genomic surveillance plays a critical role not only in identifying transmission clusters but also in characterizing the circulating *K. pneumoniae* population—an essential step in informing vaccine target selection and coverage.

The next most prevalent ST, ST2004 is much less well described. It was first defined from a 2015 Chinese isolate [58] and has publicly available genome data from China, Japan, Australia, Europe and USA, but no cases reported previously from Africa. There is insufficient data to determine the origin of this ST or the likelihood of importation from one of the countries listed above. However, the nearest neighbours in Pathogenwatch, based on LINcodes, were identified in Switzerland and the Netherlands, showing 0.32-0.64% allelic diversity. Most public ST2004 genomes include $bla_{CTX-M-15}$, as we observed in this study, suggesting this may be an emerging ESBL clone.

Profiling of AMR genes predicted most *K. pneumoniae* isolated in this study were multi-drug resistant, with 99% carrying ESBL genes and a number of STs belonging to known MDR lineages [59–62]. The majority of isolates carried acquired genes associated with resistance to aminoglycosides, ß-lactams (including ESBLs), fluoroquinolones and trimethoprim-sulfamethoxazole. No substantial change in genotypic AMR profiles was seen following the implementation of the IPC bundle (Fig 5C), which is as expected given there was no change in antimicrobial use. However, we did observe the introduction of *mphA* in ST307 clusters 2 and 4. While azithromycin is not typically used to treat neonatal sepsis, its use prophylactically to reduce maternal and neonatal postpartum infections has been studied. A recent review found modest benefits for maternal outcomes but no effect on neonatal adverse outcomes [63]. Intrapartum antibiotics may also alter the neonatal microbiome and increase AMR gene abundance, particularly in *K. pneumoniae* [64–67]. Several studies have reported elevated prescribing rates in Zambian hospitals, surpassing WHO prescribing indicators [68,69], potentially sustaining the presence of MDR pathogens within the hospital environment.

A small number of isolates carried a $bla_{NDM-5}$ carbapenemase gene, belonging to ST340 (n = 2) and ST5856 (n = 9). These isolates could not be linked to clinical metadata and so were not included in cluster analysis, however they clustered closely genetically (0–8 SNVs between ST340 isolates, 0–14 SNVs between ST5856 isolates based on Pathogenwatch clustering), indicative of some form of local transmission either within the unit or the community. Despite numbers being low, this finding is of concern given the role of carbapenems as a last line treatment option for *K. pneumoniae*. More recent data from the same hospital indicates an increase in carbapenem resistance amongst *K. pneumoniae* clinical isolates from adult patients (imipenem resistance rose from 3% in 2015, when the SPINZ study commenced, to 19% in 2020), and high rates of carbapenem resistance amongst *K. pneumoniae* isolated from the NICU environment (88%

resistant to meropenem in 2023), suggesting that carbapenem resistance has likely escalated in this setting since the conclusion of the study [70].

There are limitations to this work, primarily related to the fact that the WGS analysis was carried out retrospectively, and was not part of the original study design. This resulted in incomplete clinical data and limited linkage between laboratory and clinical datasets, ultimately constraining the scope and depth of the analysis. The presence of missing data also introduces potential biases that cannot be fully accounted for. Beyond these constraints, had the study been prospectively designed with the current analysis in mind, additional measures could have significantly enhanced the completeness and interpretability of the findings. For instance, our results suggest the elimination of some sources and persistence of others, indicated by the relatedness of strains before and after the IPC implementation. Incorporating environmental sampling during the study period could have provided direct information on the reservoirs and transmission pathways of *K. pneumoniae* on the unit, offering more specific actionable insights regarding IPC failures and guiding future intervention efforts. An additional limitation of this analysis is the exclusive use of short-read Illumina sequencing, which lacks the resolution to accurately assemble plasmids compared to Nanopore long-read data. Incorporating long-read sequencing would enhance the detection of plasmid structures and associated AMR gene content.

In conclusion, this WGS analysis of isolates collected from the SPINZ study [22] highlights the value of genomic surveillance in informing and monitoring IPC interventions in neonatal units, even when applied retrospectively. Delivery of IPC best practices in LMIC healthcare facilities may be compromised by unpredictable water supply, high patient-to-nurse ratios, facility overcrowding, personal protective equipment and hand hygiene supply shortages, reuse of single-use items, and high burden of AMR colonization and infection, among other factors [71,72]. Evidence shows that despite most neonatal units having IPC and cleaning guidelines, adequate infrastructure and consumables to support optimal IPC are lacking [73]. Furthermore the high rate of rapid-onset infections observed in this study, both pre- and post-implementation of the IPC bundle, underscores the critical importance of explicitly incorporating measures to strengthen medication safety, IV fluid handling, and surveillance for contaminated reagents, as part of IPC actions. Additionally, bacterial pathogens have adapted characteristics to enable survival on surfaces and medical equipment, such as the ability to form biofilms [74], increasing resistance to cleaning and disinfectants [75]. The WHO guidelines for IPC issued in 2016 outline eight core components for the implementation of effective IPC, however the feasibility of application depends greatly on settings, and guidelines must be adapted based on feasibility [18]. Notably, identifying and containing outbreaks, particularly in low-resource settings, presents distinct challenges due to often limited infection control training, inadequate infrastructure, and constrained healthcare resources [76]. In the NICU studied here, persistent overcrowding and a patient-to-nurse ratio of 20:1 further hinder efforts to prevent transmission among neonates.

## Supporting information

**S1 Table. Reference sequence accessions for each ST.**
(PDF)

**S2 Table. Isolates and sequence data included in the study.**
(TSV)

**S1 Fig. Pipeline of isolate from isolate extraction through to high-quality assembled genomes.** Loss of isolates from the pipeline are indicated at each stage.
(TIF)

**S2 Fig. Sensitivity analysis of ST307 clusters.** (A) Using culture date to define clusters if available, and admission date if not (n = 233), compared to (B) using only culture date (n = 206).
(TIF)

**S3 Fig. Acquired resistance genes, hypervirulence genes and birth location and onset metadata aligned to study week of infection for all other STs (of ≥4 genomes).** In all other STs for which a phylogenetic tree could not be constructed. The presence of an isolate in a cluster is indicated by the colour of the circle plotted by study week of infection. The solid vertical line represents the start of the implementation of the IPC bundle.
(TIF)

**S4 Fig. Comparison of KL102 loci in ST307.** (a) Disrupted form conserved in ST307 cluster 1 isolates from this study. (b) Wildtype reference form (GenBank accession, AB371290). Arrows show annotated protein-coding genes, navy blue arrows indicate intact open reading frames, maroon arrows indicate truncated open reading frames. Grey shows homology blocks between the two KL102 locus sequences, revealing a 661 bp deletion stretching from the end of wbaP into the beginning of wzy thus truncating both genes and presumably rendering both non-functional.
(TIF)

**S5 Fig. BLAST hits to _mph_A reference plasmid (GenBank accession: CP021752.1) in representative assembly graphs for ST307 cluster 3 and 4 genomes.** BLAST search and visualisation of the graph was done using Bandage. In each graph, rectangles indicate contigs, sized to indicate their relative length in base pairs. Contigs are coloured to indicate hits to the reference plasmid (GenBank accession CP021752.1), using a rainbow scale such that red indicates hits to the start of the reference plasmid sequence, colours yellow through green indicate hits in the middle of the reference sequence, through to blue for hits at the end of the reference sequence. The contig containing the _mph_A gene is labelled. (A) Cluster 4 genome, read accession ERR15165518. Zoomed in on the part of the graph with hits to the reference plasmid (96.9% coverage at mean 98.6% identity), note this part is connected to the rest of the graph including the chromosome. (B) Cluster 3 genome, ZN2356 (accession ERR15165559). Note the part of the graph shown is disconnected from the rest of the graph including the chromosome. Coverage of the reference plasmid is 74%.
(TIF)

**S6 Fig. Maximum likelihood phylogenetic tree and pairwise SNV distance matrix of a subset of ST101 infections.** Tree tips labelled with cluster number (colour) and week of infection. Scale bar indicates genetic distance (SNVs). Two infections observed in week 6 during the baseline period (cluster 1, green tree tips), and four infections during weeks 39–44 in the post-implementation period (cluster 2, purple tree tips) as well as a single non-clustered infection (black tree tip) at week 48.
(TIF)

**S1 Appendix. Clustering sensitivity analyses.** The sensitivity of clustering to choice of SNV threshold (between 5–25) and temporal threshold (between 7–28 days) was assessed for all STs.
(DOCX)

## Acknowledgments

We thank the participants and their families in the SPINZ study, and the clinical and laboratory staff involved in collection and processing of relevant data and isolates. We also thank Andrew Page, David Baker, Leonardo de Oliveira Martins and the Core Sequencing and Bioinformatics teams at the Quadram Institute (QIB), Norwich, United Kingdom for their assistance with Illumina sequencing. This work was supported by Monash eResearch capabilities, including M3. We thank the Institut Pasteur teams for the curation and maintenance of BIGSdb-Pasteur databases at http://bigsdb.pasteur.fr/.

## Author contributions

**Conceptualization:** Susan E. Coffin, Kathryn E. Holt, Davidson H. Hamer.

**Data curation:** Matthew Bates.

**Formal analysis:** Laura T. Phillips, Kathryn E. Holt.

**Funding acquisition:** Susan E. Coffin, Kathryn E. Holt, Davidson H. Hamer.

**Investigation:** Matthew Bates, Ebenezer Foster-Nyarko, Monica Kapasa, Sylvia Machona, Lawrence Mwananyanda, James C. L. Mwansa, Chileshe L. Musyani, John M. Tembo, Franklyn N. Egbe.

**Resources:** Susan E. Coffin, Davidson H. Hamer.

**Visualization:** Laura T. Phillips, Kathryn E. Holt.

**Writing – original draft:** Laura T. Phillips, Kathryn E. Holt.

**Writing – review & editing:** Laura T. Phillips, Matthew Bates, Susan E. Coffin, Ebenezer Foster-Nyarko, Kathryn E. Holt, Davidson H. Hamer.

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
