## [Decision Letter · Decision Letter 0]

21 Jan 2026

PGPH-D-25-03829

Transmission dynamics of *Klebsiella pneumoniae* in a neonatal intensive care unit in Zambia before and after an infection control bundle

Dear Dr. Holt,

Thank you for submitting your manuscript to PLOS Global Public Health. After careful consideration, we feel that it has merit but does not fully meet PLOS Global Public Health’s publication criteria as it currently stands. Therefore, we invite you to submit a revised version of the manuscript that addresses the points raised during the review process.

We look forward to receiving your revised manuscript.

Kind regards,

Maya Nadimpalli, PhD

Academic Editor

Journal Requirements:

1. Please provide a detailed online Financial Disclosure statement. This is published with the article. It must therefore be completed in full sentences and contain the exact wording you wish to be published.

a) Please clarify all sources of financial support for your study. List the grants, grant numbers, and organizations that funded your study, including funding received from your institution. Please note that suppliers of material support, including research materials, should be recognized in the Acknowledgements section rather than in the Financial Disclosure.

b) State the initials, alongside each funding source, of each author to receive each grant. For example: “This work was supported by the National Institutes of Health (####### to AM; ###### to CJ) and the National Science Foundation (###### to AM).”

c) State what role the funders took in the study. If the funders had no role in your study, please state: “The funders had no role in study design, data collection and analysis, decision to publish, or preparation of the manuscript.”

For more information, please go to our submission guidelines:

https://journals.plos.org/globalpublichealth/s/submission-guidelines#loc-financial-disclosure-statement

2. Please ensure that the funders and grant numbers match between the Financial Disclosure field and the Funding Information tab in your submission form. Note that the funders must be provided in the same order in both places as well.

3. Please upload your main article file as a .doc, .docx or .rtf file.

4. Please provide separate main figure files in .tif or .eps format only and remove any figures embedded in your manuscript file. Please also ensure that all files are under our size limit of 10MB. Please leave the figure captions in the manuscript.

5. We have noticed that you have uploaded Supporting Information files, but you have not included a list of legends. Please add a full list of legends for your Supporting Information files before or after the references list.

Additional Editor Comments (if provided):

Reviewers' comments:

Reviewer's Responses to Questions

**Comments to the Author**

1. Does this manuscript meet PLOS Global Public Health’s publication criteria?

Reviewer #1: Yes

Reviewer #2: Yes

2. Has the statistical analysis been performed appropriately and rigorously?

Reviewer #1: Yes

Reviewer #2: Yes

3. Have the authors made all data underlying the findings in their manuscript fully available (please refer to the Data Availability Statement at the start of the manuscript PDF file)?

Reviewer #1: Yes

Reviewer #2: Yes

4. Is the manuscript presented in an intelligible fashion and written in standard English?

Reviewer #1: Yes

Reviewer #2: Yes

Reviewer #1: This is a fascinating paper presenting a highly robust genomic analysis of a large collection of Klebsiella pneumoniae isolates. Although the study is limited by the absence of contemporaneous outbreak investigations and environmental sampling to more definitively delineate transmission pathways, the authors effectively triangulate genomic, temporal, and clinical data to generate plausible inferences regarding transmission dynamics and the hypothesized impact of the IPC bundle.

Minor Comments

1. The opening paragraph would benefit from clarification of denominators to avoid confusion. In the second sentence, I suggest explicitly adding “sepsis-related” to clarify the statistic, e.g. “93.9% of global sepsis-related neonatal deaths in 2019”. As currently written, the transition between infectious neonatal deaths overall and neonatal sepsis specifically is potentially confusing.

2. The statement that “this is the first study to perform genomic characterization using WGS, including sequence typing and AMR gene profiling, to explore the prevalence and distribution of STs” appears overly broad. Do the authors mean this is the first such study from Zambia?

3. The observation that the dominant outbreak clone (ST307) was predicted to be “capsule-null” is fascinating, particularly given that capsule-null K. pneumoniae are rarely recovered from blood due to the capsule’s role in evading serum complement and phagocytosis. The authors reasonably speculate that these isolates may in fact have been encapsulated via an alternative locus, given the scale and persistence of invasive disease. However, an additional epidemiologic explanation deserves consideration: An alternative hypothesis is that these organisms may not have required particularly strong immune evasion traits because they were introduced directly into the bloodstream—for example via contaminated intravenous fluids, reagents, or equipment—rather than via a colonization-preceding translocation pathway. This interpretation aligns with later discussion of rapid-onset cases and would be consistent with sustained transmission from a persistent environmental reservoir rather than enhanced intrinsic virulence. The authors go on to conclude that this finding “underscores the importance of considering both O- and K-antigen variability in the design of a maternal vaccine.” I agree, but would go further: these data also underscore the importance of real-time epidemiologic investigation and environmental source tracing, as this clone may not have been uniquely virulent but rather uniquely positioned to persist in, and be transmitted from, the NICU environment through breaches in medication safety or IPC. These clones, due to their high numbers, are likely making the “cut” for a maternal vaccine, but they may in fact not be very virulent, and may simply be reflective of their persistence in the environment of that NICU.

4. The authors should clarify how repeat blood cultures from the same patient were handled. Were multiple cultures obtained per neonate, and if so, what criteria (time interval, clinical resolution) were used to define a new episode versus persistence of the same infection?

5. Given the authors’ own acknowledgment that limited availability of blood cultures constrains accurate diagnosis and outbreak detection in LMIC NICUs, it would be helpful to explicitly state whether blood culture bottles were consistently available throughout the entire surveillance period. If access was uninterrupted, this should be clearly stated, as it strengthens confidence that observed temporal trends reflect true changes in transmission rather than diagnostic availability.

6. The authors’ discussion of the limitations of conventional HCAI definitions—particularly the EOS vs. LOS dichotomy and the routine attribution of EOS to vertical transmission—is a notable strength of this paper. The genomic evidence convincingly demonstrates that many “early-onset” cases are compatible with nosocomial transmission, challenging entrenched assumptions in neonatal epidemiology.

7. Please spell out “IV” as “intravenous” at first mention.

8. The paper does an admirable job of interrogating transmission pathways and the impact of the IPC bundle using genomic data and limited clinical data alone. The overall reduction in bloodstream infections is compelling. However, one unavoidable limitation is the possibility that the pre-intervention period coincided with an anomalously large outbreak. While the authors appropriately acknowledge this indirectly, readers will naturally wonder whether reduced transmission was sustained beyond the study period. Understanding that this is beyond the scope of this analysis, the authors attempted to look at the bundle’s efficacy from a different angle: by examining rapid- versus delayed-onset infections, hypothesizing that a decolonization-focused bundle would preferentially reduce delayed-onset sepsis. They write, “3 out of 6 clusters post-implementation included at least half rapid-onset infections”, which is factually correct, but the aggregate data (Table 2) show that 44% (82/187) were rapid onset in the baseline period, and 37% (28/76) were rapid onset in the post-implementation period. One might have expected a decline in delayed-onset cases if decolonization was the dominant mechanism of effect and the aggregate data don't reflect this. This discrepancy does not undermine the findings, but it does suggest that the observed benefit may reflect a broader unit-wide IPC ‘push’ rather than the specific mechanistic effects of individual bundle components. This distinction is important, as it informs how future IPC bundles should be designed and evaluated. It may bear mentioning, given the sheer magnitude of rapid-onset infections in this analysis, that future IPC bundles should explicitly incorporate interventions aimed at preventing rapid-onset sepsis, including strengthened medication safety, IV fluid handling, and surveillance for contaminated reagents. These elements were not clearly part of the intervention described here (unless embedded within IPC training), and their omission may explain why rapid-onset transmission persisted despite overall gains.

Reviewer #2: Phillips et al. present a compelling study investigating whole-genome sequencing (WGS) data from over 400 Klebsiella pneumoniae isolates collected from neonates at a large tertiary care hospital in Zambia. The WGS data derive from a longitudinal study design consisting of a 6-month baseline period, followed by a 6-week infection prevention and control (IPC) intervention and an 11-month post-intervention assessment. Genomic analysis revealed that nearly all isolates carried extended-spectrum β-lactamase (ESBL) genes and belonged to the globally disseminated ST307 lineage. Further analyses demonstrated that ST307 was responsible for widespread transmission during the baseline period and that IPC measures effectively halted circulation of this outbreak lineage. WGS also identified the emergence of a genetically distinct ST307 strain post-intervention, suggesting clearance of the original strain and subsequent introduction of a new one. Similar transmission dynamics were observed in other major lineages, underscoring the effectiveness of IPC practices in interrupting outbreaks. Notably, baseline ST307 isolates exhibited disruption of the KL102 locus, raising the possibility that capsule alteration may have conferred an in-host advantage, such as enhanced biofilm formation. Overall, this study provides a strong example of how integrated IPC strategies and WGS surveillance can be used to intervene on high-risk pathogens. The bioinformatic methods are sound, and the conclusions are well supported by the results. Therefore, only a few minor edits are recommended.

Minor Comments

1. Please add scale bars to all phylogenetic trees to facilitate interpretation of genetic distances.

2. Consider moving lines 746–747 from the Discussion section to line 315 in the Results section, as this content appears more appropriate for reporting primary findings.

**Do you want your identity to be public for this peer review?** For information about this choice, including consent withdrawal, please see our Privacy Policy

Reviewer #1: **Yes:** Jonathan Strysko

Reviewer #2: No

---

## [Editor Report · Decision Letter 1]

27 Jan 2026

Transmission dynamics of *Klebsiella pneumoniae* in a neonatal intensive care unit in Zambia before and after an infection control bundle

PGPH-D-25-03829R1

Dear Dr. Holt,

We are pleased to inform you that your manuscript 'Transmission dynamics of *Klebsiella pneumoniae* in a neonatal intensive care unit in Zambia before and after an infection control bundle' has been provisionally accepted for publication in PLOS Global Public Health.

Best regards,

Maya Nadimpalli, PhD

Academic Editor